# PSTPIP1-Associated Myeloid-Related Proteinemia Inflammatory (PAMI) Syndrome: A Systematic Review

**DOI:** 10.3390/genes14081655

**Published:** 2023-08-19

**Authors:** Manel Mejbri, Raffaele Renella, Fabio Candotti, Cecile Jaques, Dirk Holzinger, Michael Hofer, Katerina Theodoropoulou

**Affiliations:** 1Pediatric Immuno-Rheumatology of Western Switzerland, Department Women-Mother-Child, Lausanne University Hospital and University of Lausanne, 1011 Lausanne, Switzerland; manel.mejbri@hcuge.ch (M.M.); michael.hofer@chuv.ch (M.H.); 2Pediatric Hematology-Oncology Unit, Division of Pediatrics, Department Women-Mother-Child, Lausanne University Hospital, 1011 Lausanne, Switzerland; raffaele.renella@chuv.ch; 3Division of Immunology and Allergy, Lausanne University Hospital and University of Lausanne, 1011 Lausanne, Switzerland; fabio.candotti@chuv.ch; 4Medical Library, Lausanne University Hospital and University of Lausanne, 1011 Lausanne, Switzerland; cecile.jaques@chuv.ch; 5Department of Pediatric Hematology-Oncology, University of Duisburg-Essen, 45147 Essen, Germany; dirk.holzinger@uk-essen.de; 6Department of Applied Health Sciences, University of Applied Sciences Bochum, 44801 Bochum, Germany

**Keywords:** *PSTPIP1* gene, PAMI syndrome, E250K, E257K, Hyperzincemia/Hypercalprotectinemia, cytopenia, autoinflammation

## Abstract

PSTPIP1 (proline-serine-threonine phosphatase-interactive protein 1)-associated myeloid-related proteinemia inflammatory (PAMI) syndrome, previously known as Hyperzincemia/Hypercalprotectinemia (Hz/Hc) syndrome, is a recently described, rare auto-inflammatory disorder caused by specific deleterious variants in the *PSTPIP1* gene (p.E250K and p.E257K). The disease is characterized by chronic systemic inflammation, cutaneous and osteoarticular manifestations, hepatosplenomegaly, anemia, and neutropenia. Increased blood levels of MRP 8/14 and zinc distinguish this condition from other PSTPIP1-associated inflammatory diseases (PAID). The aim of this systematic review is to provide a comprehensive overview of the disease phenotype, course, treatment, and outcome based on reported cases. This systematic review adheres to the PRISMA guidelines (2020) for reporting. A literature search was performed in Embase, Medline, and Web of Science on 13 October 2022. The quality of the case reports and case series was assessed using the JBI checklists. Out of the 43 included patients with PAMI syndrome, there were 24 females and 19 males. The median age at onset was 3.9 years. The main clinical manifestations included anemia (100%), neutropenia (98%), cutaneous manifestations (74%), osteoarticular manifestations (72%), splenomegaly (70%), growth failure (57%), fever (51%), hepatomegaly (56%), and lymphadenopathy (39%). Systemic inflammation was described in all patients. Marked elevation of zinc and MRP 8/14 blood levels were observed in all tested patients. Response to treatment varied and no consistently effective therapy was identified. The most common therapeutic options were corticosteroids (N = 30), anakinra (N = 13), cyclosporine A (N = 11), canakinumab (N = 6), and anti-TNF (N = 14). Hematopoietic stem cell transplantation has been recently reported to be successful in five patients. Our review highlights the key characteristics of PAMI syndrome and the importance of considering this disease in the differential diagnosis of patients presenting with early-onset systemic inflammation and cytopenia.

## 1. Introduction

PSTPIP1 (proline-serine-threonine phosphatase-interactive protein 1)-associated myeloid-related proteinemia inflammatory (PAMI) syndrome, previously known as Hyperzincemia/Hypercalprotectinemia (Hz/Hc) syndrome, is a recently described, rare autoinflammatory disorder caused by specific deleterious variants in the *PSTPIP1* gene (p.E250K and p.E257K) [1,2,3].

The clinical and biological phenotype of PAMI syndrome differentiates this condition from PAPA (pyogenic arthritis, pyoderma gangrenosum [PG], acne) syndrome and the other PSTPIP1-associated inflammatory diseases (PAID). Patients with PAMI syndrome present with chronic systemic inflammation, clinical features of osteoarticular and/or cutaneous manifestations, lymphoproliferation, anemia, and neutropenia. A hallmark of the disease are the increased serum levels of myeloid-related protein (MRP) 8/14 (S100A8/A9 or calprotectin) and zinc [3].

The pathophysiological mechanisms of PAMI syndrome are not completely elucidated. PSTPIP1 is a regulatory phosphatase protein that modulates T-cell and phagocyte activation, cytoskeletal organization, and IL-1 production [1]. Specific *PSTPIP1* deleterious variants causing PAMI (p.E250K and p.E257K) are thought to lead to an increased binding to the pyrin inflammasome, resulting in inflammasome activation and interleukin (IL)-1β overproduction [1,3].

Although progress has been made in its clinical and genetic characterization, PAMI syndrome is a challenging disease to diagnose because of its rarity, heterogeneity, and overlap with other autoinflammatory conditions. Moreover, clinical management of the disease remains arduous. Various therapeutic approaches have been explored, including non-steroidal anti-inflammatory drugs (NSAIDs), corticosteroids, conventional disease-modifying anti-rheumatic drugs (cDMARDs), and biological disease-modifying anti-rheumatic drugs (bDMARDs), with no effective standard therapy established to date.

Our review aims to provide a comprehensive overview of the disease phenotype, course, treatment, and outcome based on reported cases, and discusses the current understanding of this poorly recognized disease.

## 2. Methods

### 2.1. Declaration and Protocol

This systematic review is reported following the PRISMA (Preferred Reporting Items for Systematic reviews and Meta-Analyses) guidelines 2020 [4].

The review protocol was published in the PROSPERO registry database (CRD42022376098).

### 2.2. Inclusion and Exclusion Criteria

#### 2.2.1. Inclusion Criteria

##### Types of Studies

We included all types of studies (e.g., case reports, case series, prospective observational studies). Only human research and articles with full text available in the English language were included.

##### Types of Participants

Patients with any of the following characteristics were included:Confirmed PAMI syndrome in the presence of one of the PAMI-specific *PSTPIP1* gene deleterious variants (E250K or E257K).Confirmed Hyperzinceamia and Hypercalprotectineamia syndrome in the presence of one of the PAMI-specific *PSTPIP1* gene deleterious variants (E250K or E257K).Clinical features of PAPA syndrome in the presence of one of the PAMI-specific *PSTPIP1* gene deleterious variants (E250K or E257K).Clinical features of PAID in the presence of one of the PAMI-specific *PSTPIP1* gene deleterious variants (E250K or E257K).

#### 2.2.2. Exclusion Criteria

Patients with of one of the PAMI-specific *PSTPIP1* gene deleterious variants (E250K or E257K), but without any clinical or biological manifestations.Patients reported as carrying the clinical diagnosis PAMI syndrome, but without evidence of PAMI-specific *PSTPIP1* gene deleterious variants (E250K or E257K) (i.e., genetic analysis not performed, or alternative deleterious variants identified).

It should be noted that the references by Sampson, Isidor, Fessatou, Sugiura and Demidowich et al. [5,6,7,8,9] were initially excluded, but later included in relation to the genetic results published in the study by Holzinger et al. in 2015 [3].

### 2.3. Search Strategy and Selection Process

A literature search was performed using three Databases (Embase.com, Medline All Ovid and Web of Science Core collection) on 13 October 2022 by a researcher (MM) with the assistance of a medical librarian (CJ). There were no limits regarding the research strategies. The complete search strategies used are shown in Appendix A.

Retrieved records were imported into Endnote 20 (Clarivate Analytics, USA) and duplicates were removed (CJ). Then, MM and KT proceeded to the selection of articles, working independently and in a double-blind way using the Rayyan Web tool [10]. The titles and abstracts of all remaining articles were reviewed, and those that were not relevant to the review topic or did not meet the inclusion criteria were excluded. The same two researchers conducted a full reading of the texts and then resolved disagreements through consensus. Finally, the quality of the case reports and case series was assessed by Manel Mejbri using the JBI checklists [11,12]. The JBI checklist for the case reports consisted of 8 items with a classification of ‘yes’, ‘no’, ‘unclear’, or ‘not applicable’. Items 7 and 8 were not applicable for our review. Only cases with ‘yes’ items [4,5,6] were included. The JBI checklist for case series consisted of 10 items with a classification of ‘yes’, ‘no’, ‘unclear’, or ‘not applicable’. Only cases with ‘yes’ items [7,8,9,10] were included. A PRISMA flow diagram describing the case selection process is shown in Figure 1 [4].

### 2.4. Data Collection and Data Items

An Excel spreadsheet was used to manage and extract relevant data from the articles included in the review. To test for accuracy, a second researcher reviewed extracted data for at least 50% of the papers. The spreadsheet included details such as publication details (year, author, journal), age of onset, sex, family history, general signs, lymphadenopathy/hepatomegaly/splenomegaly, dermatologic signs, osteoarticular signs, hematologic signs, growth failure, other clinical symptoms reported, biologic findings (WBC, ANC, Hb, platelets, CRP, SAA, ESR, plasma zinc, MRP8/14), genetic results, and previous and current treatment (dosage, duration, and adverse events), as well as outcome and evolution, if available.

If the information was not available in the article, it was noted as such.

## 3. Results

Our review identified 252 articles through database searches. After eliminating duplicates and excluding unqualified articles, we included 25 relevant articles that described 43 patients with PAMI syndrome [1,2,3,5,6,7,8,9,13,14,15,16,17,18,19,20,21,22,23,24,25,26,27,28,29,30].

The demographics and the clinical presentations of the patients are summarized in Table 1.

### 3.1. Demographics

Out of the 43 included patients with PAMI syndrome, 24 were females (56%) and 19 were males (44%), resulting in a male: female ratio of 0.79. Unfortunately, we were unable to determine any clustering of ethnic groups or pattern of consanguinity in these cases, as many articles did not provide this information.

We observed a median age at disease onset to be 3.9 years. In 38% of patients, the first symptoms appeared before their first birthday. In the reported cases, the disease can manifest at any time between birth and 18 years of age; however, over 80% of patients experienced their first symptoms before the age of six years (Figure 2). The age at diagnosis varied widely, ranging from 6 months to 56 years. It is important to highlight that there was often a significant delay from clinical presentation to diagnosis, with a median delay of 9.3 years.

### 3.2. Clinical Presentation and Disease Course

Patients with PAMI syndrome presented with early-onset systemic inflammation, along with a variety of inflammatory features ranging from mild symptoms to severe presentations. In our review a predominant pattern emerged, with 88% of cases exhibiting serious manifestations characterized by severe cytopenia and/or prominent cutaneous or osteoarticular symptoms, and in some cases growth failure and organ involvement. A milder phenotype was observed in 12% of cases, characterized by biologic inflammation with mild cytopenia and/or mild skin or osteoarticular symptoms.

Interestingly, phenotypic heterogeneity was observed among family members sharing the same variants [2,3,19,21,29]. In most patients, clinical symptoms appeared gradually over a period of several years.

Osteoarticular manifestations were described in 72% (31/43) of cases, while cutaneous manifestations were seen in 74% (32/43). Splenomegaly complicated the presentation of 70% of patients (29/41), whereas hepatomegaly was seen in 56% (23/41). Growth failure was reported in 57% (19/33), and fevers were described in 51% of patients (19/37). Less frequently, lymphadenopathy (39%), gastrointestinal manifestations (16%), renal involvement (13%), hemorrhagic diathesis with recurrent epistaxis and/or hematoma tendency (11%), and developmental delay (11%) were reported. Severe pulmonary arterial hypertension (HTAP) with heart failure was described in one patient.

Furthermore, 16 (37%) out of 43 patients were reported to have experienced recurrent infections, two of which were complicated by septic shock. One of these two patients died at 36 years of age due to multiple organ failure. Macrophage activation syndrome (MAS) was reported in 4 patients (9%).

The main clinical manifestations of PAMI syndrome and their frequency are illustrated in Figure 3 and Figure 4.

### 3.3. Laboratory, Histology and Genetic Findings

All patients had hematological involvement, i.e., mild to severe anemia in 100% and neutropenia in 98% of cases, respectively (with ranges of Hb = 20–117 g/L and ANC = 0.08 to 1.3 G/L). Thrombocytopenia was present in 41% of cases, while 9% exhibited thrombocytosis.

Systemic inflammation was evidenced in all reported cases using the monitoring of CRP, ESR or SAA. Markedly elevated concentrations of zinc and MRP8/14 were detected in all tested patients.

Immunologic investigations revealed the presence of antineutrophil antibodies or laboratory data consistent with autoimmune neutropenia in 53% of the patients studied (8/15). The presence of antinuclear antibodies and rheumatoid factor was reported, respectively, in 13 and 9 patients. Among them, only one patient tested positive for both. This patient also exhibited decreased complement levels (C3, C4) and a positive direct antiglobulin test.

NK cell deficiency was observed in 7 out of 12 studied patients (58%). Additionally, cytokine profiling performed in 15 cases revealed elevated levels of IL-18 in all tested patients (12 out of 12) and high levels of IL-1β in 13 out of 14 cases, with the only case presenting normal IL-1β levels undergoing treatment. Furthermore, an increased expression of type I interferon (IFN)-regulated genes (positive IFN signature) was reported in 3 cases.

Bone marrow aspiration and/or biopsy was performed in 23 patients. Among these, normal bone marrow cellularity was found in only 3 patients. Pathologic findings ranged from hypercellularity (4/23) to bone marrow hypoplasia (6/23), different degrees of fibrosis (3/23) and mono- or multilinear dysplasia (10/23), with dyserythropoiesis found in most of the patients (7/23) followed by dysmyelopoiesis (3/23) and tri-lineage dysplasia (2/23).

Genetic analysis of the *PSTPIP1* gene revealed that in a significant majority of cases, 41 out of 43 (95%) exhibited the heterozygous E250K variant. Only two patients in our review had the E257K variant. Among the reviewed cases, 6 out of 26 involved an inheritance from one of their parents, while 20 cases (76%) were carriers of the de novo variants. Unfortunately, information was unavailable for 17 patients.

### 3.4. Treatment and Outcome

In the reported cases, various therapeutic approaches were used, including NSAIDs (8), systemic corticosteroids (n = 30), anti-IL1 (anakinra (n = 13), canakinumab (n = 6)), anti-TNF (adalimumab (n = 3), etanercept (n = 6), infliximab (n = 7)), cyclosporine A (n = 11), tocilizumab (n = 4), methotrexate (n = 4), tacrolimus (n = 3), and colchicine (n = 8). However, response to treatment was extremely variable (Figure 5), with no consistently effective therapy identified. In cases where these therapies were effective, the authors observed a reduction or resolution of systemic inflammation, osteoarticular manifestations, skin lesions, improvement in anemia, and, to a lesser degree, improvement in hepatosplenomegaly and growth. Neutropenia persisted in all patients independently of the treatment used. In six patients, the administration of granulocyte colony-stimulating factor (G-CSF) was reported.

Notably, among the patients who received anti-IL1 therapy, 40% (6/15) showed significant improvement, either when given as monotherapy (4/6), or in combination therapy with cyclosporine (1/6) or tacrolimus and oral steroids (1/6). On the other hand, anti-TNF treatment yielded variable results, with 57% (8/14) of patients showing improvement of osteoarticular and/or cutaneous manifestations and a reduction in biologic inflammation. Half of the patients required a combination treatment; two with steroids, one with methotrexate, and one with methotrexate, cyclosporine and steroids. Tocilizumab was found to be ineffective overall, as only one patient reported an improvement before discontinuing treatment due to lack of persistent efficacy. Cyclosporine A demonstrated partial effectiveness in 9 out of 11 patients, however it was administered in combination with steroids in all cases, with or without another biologic treatment. Interestingly, NSAIDs as monotherapy were found to be effective in addressing systemic inflammation, anemia, and osteoarticular symptoms in three patients.

In 2020, Laberko et al. [20] became the first team to conduct an allogenic hematopoietic stem cell transplantation (HSCT) for five patients with PAMI syndrome. In four of these patients, the indication was lack of PAMI disease control, while the occurrence of myelodysplastic syndrome justified HSCT in the fifth case. HSCT was successful in four cases, while a fifth patient required a second HSCT due to graft rejection. At the last follow-up (median 2.2 years), all five patients were free of PAMI symptoms, with adequate immune recovery.

## 4. Discussion

This review describes the phenotype of 43 patients with confirmed PAMI syndrome. PAMI syndrome is characterized by early onset systemic inflammation, cytopenia, lymphoproliferation, cutaneous and/or osteoarticular manifestations. While primarily affecting these areas, our review highlighted that other organs may also be affected considerably. Several authors have reported gastrointestinal involvement, with 16% of cases experiencing flare-ups of colitis and/or chronic diarrhea [3,16,19,20,24,28,29]. Notably, 57% of these patients developed the disease within the first few months of life, suggesting that PAMI syndrome should be considered in the differential diagnosis of early onset inflammatory bowel disease (IBD). Similarly, kidney involvement has been identified in PAMI syndrome, as indicated by Borgia et al. [23]. They found that heterogenous nephropathies can be part of the clinical spectrum of the syndrome, underlining the importance of regular renal evaluations. The same authors suggested canakinumab as a possible treatment for these manifestations. Another significant aspect of PAMI syndrome is neurologic impairment, with 11% of cases exhibiting developmental delay. Del Borello et al. recently reported a case showing significant improvement in the neurobehavioral profile and resolution of cerebral atrophy six months after initiating anakinra treatment [22]. These findings highlight the broad spectrum of clinical manifestations associated with PAMI syndrome, which can vary from mild cases without skin and/or osteoarticular manifestations to severe cases with a progressive course and organ involvement. Interestingly, even within the same family with an identical variant, the phenotype showed variability, probably due to an incomplete penetrance [29], as also reported in PAPA syndrome. Despite the distinct clinical presentations and the phenotypic variability, our review affirms that cytopenia, systemic inflammation, and elevated levels of zinc and MRP8/14 consistently emerge as the hallmark features observed in all patients. These markers may be used as screening tests for suspected cases in order to facilitate prompt diagnosis and an earlier disease control.

The pathophysiology of hematologic findings in PAMI syndrome remains unclear. Interestingly, neutropenia remained unchanged in all patients under treatment, despite the improvement of anemia and the resolution of systemic inflammation reported in some cases.

Interestingly, among the 43 PAMI cases reported in this review, one presented with a lupus-like phenotype (patient N° 8, Table 1) [17,31], suggesting that including the *PSTPIP1* gene in lupus genetic panels might be of interest. Broadening the scope of genetic testing in patients presenting with the PAMI phenotype may lead to the discovery of new PAMI-specific variants in the future. Recently, a novel *PSTP1* variant (N236K) has been reported to be associated with the PAMI phenotype, suggesting that E250K and E257K may not be the only PAMI-specific *PSTPIP1* gene deleterious variants [32].

No consistently effective therapy has yet been identified for PAMI syndrome. The response to anti-cytokine therapies varies among cases, with some patients demonstrating significant improvement upon initiation of such treatments, while others show no response. Globally, anti-TNF therapy appears to be more effective than anti-IL-1 therapy, with 57% of patients experiencing improvement compared to 40% with anti-IL-1 therapy. These improvements include skin and osteoarticular manifestations, as well as systemic inflammation, lymphoproliferation, and anemia. However, the efficacy of anti-TNF agents was more often observed in a combination treatment with steroids and/or cDMARDs, while anti-IL-1 therapy seems to be more frequently effective as a monotherapy. Notably, anti-IL-1 therapy has shown promising efficacy even for severe manifestations involving the neurologic, gastrointestinal and renal systems [19,22,23]. Our review also suggests that there may also be a role for NSAID, corticosteroids, or cyclosporine in the symptomatic treatment of this condition.

HSCT demonstrated successful outcomes in a small number of patients treated to date, leading to the resolution of PAMI symptoms, normalization of inflammatory markers and zinc levels, and discontinuation of specific therapies. This is in line with the prevailing view that the effectiveness of HSCT in autoinflammatory diseases is still uncertain. HSCT could be considered for patients with an incomplete response to conventional therapies, including those requiring G-CSF therapy. However, further research with larger patient numbers and longer follow-up is essential to determine whether HSCT may provide a curative option for a subset of PAMI patients with a severe disease course.

This work is subject to certain limitations due to its retrospective design and other biases associated with the selection criteria of patients. We excluded patients who did not have one of the PAMI-specific *PSTPIP1* gene deleterious variants (E250K or E257K), as well as cases who possessed one of these variants but lacked clinical manifestations. Although these criteria were implemented to mitigate the risk of bias, they resulted in a smaller cohort for analysis.

In conclusion, our review highlights the key characteristics of PAMI syndrome, including early onset systemic inflammation, cytopenia, lymphoproliferation, and cutaneous and/or osteoarticular manifestations. The presence of other disease features and the course of the disease exhibited significant variability, often leading to high morbidity. We emphasize the importance of considering PAMI syndrome in the differential diagnosis of patients presenting with early-onset systemic inflammation and cytopenia. Measuring serum levels of zinc and MRP 8/14 in these patients may facilitate early diagnosis and treatment, potentially improving patients’ quality of life and disease outcome.

## Figures and Tables

**Figure 1 genes-14-01655-f001:**
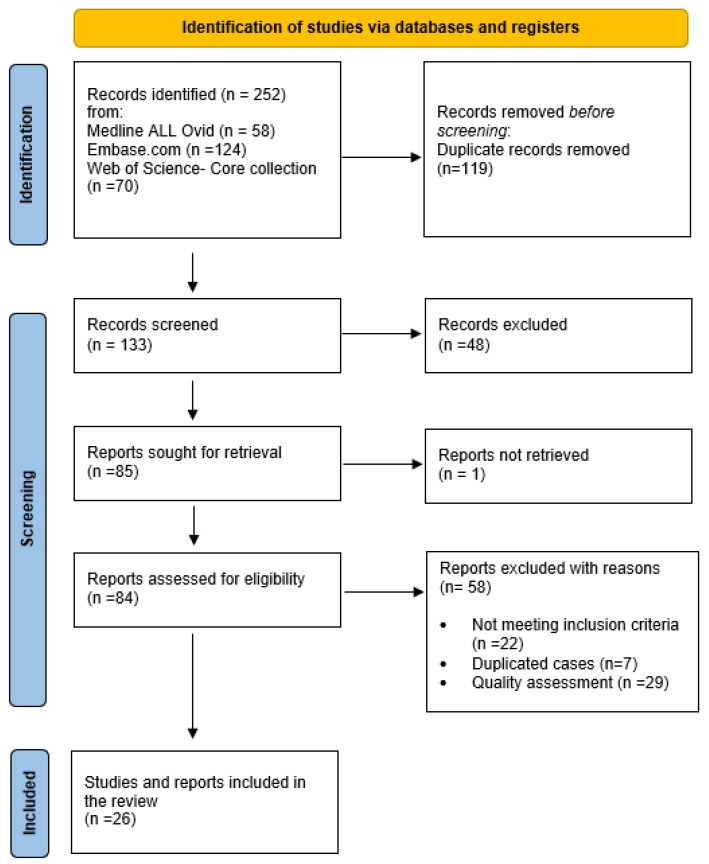
Flow diagram.

**Figure 2 genes-14-01655-f002:**
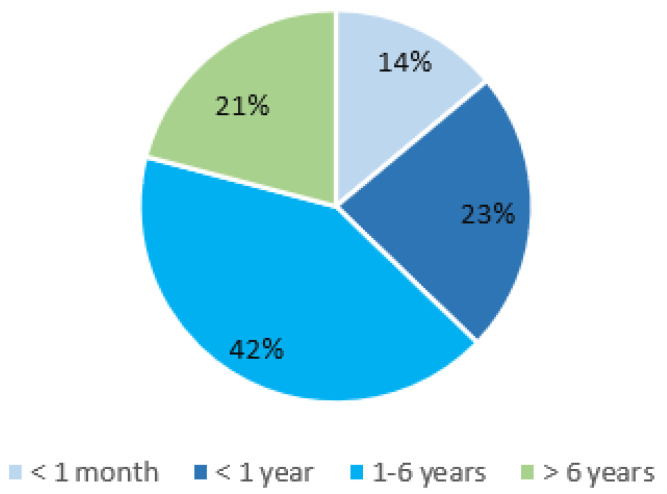
PAMI syndrome: age of onset.

**Figure 3 genes-14-01655-f003:**
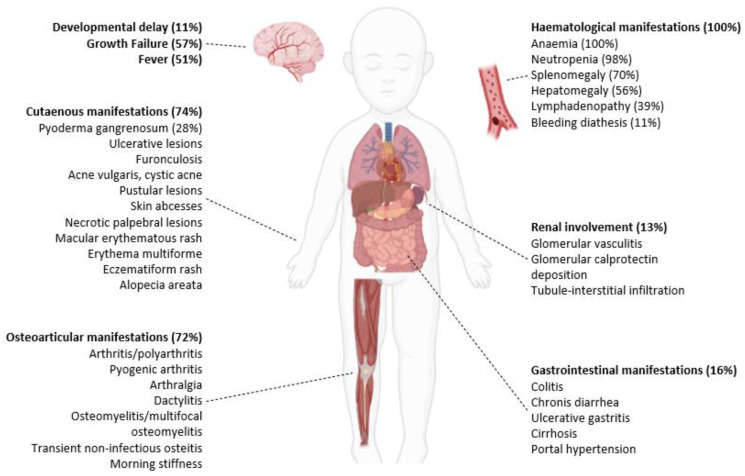
Clinical manifestations of PAMI syndrome. BioRENDER illustration.

**Figure 4 genes-14-01655-f004:**
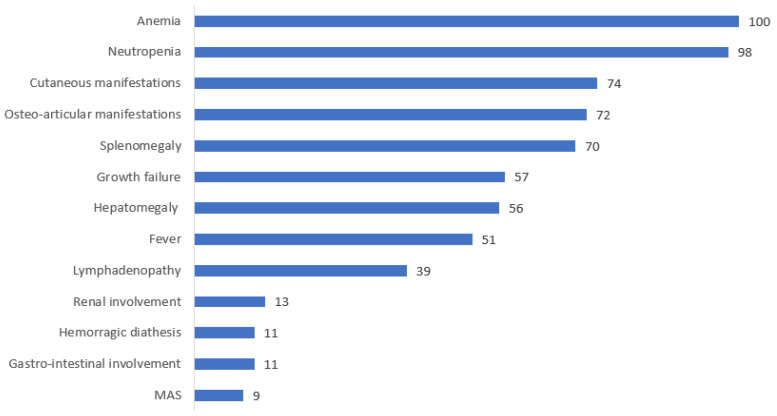
Distribution of the clinical manifestations of PAMI syndrome.

**Figure 5 genes-14-01655-f005:**
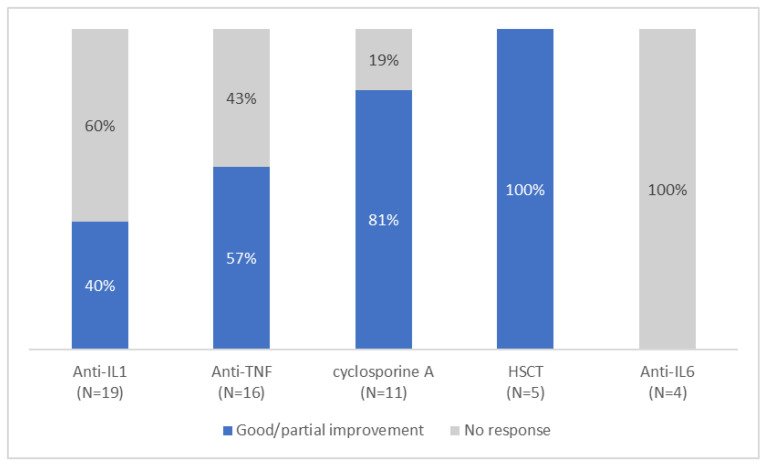
Response to different treatment.

**Table 1 genes-14-01655-t001:** Clinical features of patients with PAMI syndrome.

N°	Author	Age of Onset	Age in Years	Sex	Family History	Fever	Growth Failure	Lymphadenopathy Hepatosplenomegaly	Dermatologic Signs	Osteoarticular Signs	Hematologic Signs	Other Symptoms
1	Zheng et al., 2022 [14]	6 months	6	F	Chinese, healthy parents	Yes	NA	Hepatosplenomegaly	No	Recurrent pyogenic arthritis	Anemia Neutropenia MAS	No
2	Zheng et al., 2022 [14]	3 years	7	F	NA	NA	NA	Splenomegaly	Skin ulceration	No	Anemia Pancytopenia	NA
3	Whiteside et al., 2022 [15]	Neonatal period	3	F	African descent and healthy parents	No	No	No	No	No	Neutropenia	Seizures, speech, and cognitive delay secondary to bilateral infarction in the neonatal period with history of fetal distress, high inflammatory markers
4	Cox et al., 2022 [16]	2 years	46	M	Irish, nonconsanguineous healthy parent	No	Yes	Lymphadenopathy	Nodulo-cystic acne	Intermittent synovitis and arthritis	Thrombocytopenia Neutropenia	Flare of colitis, recurrent lower respiratory tract infections
5	Cox et al., 2022 [16]	6 months	7	M	Irish, non-consanguineous parents, mother with arthritis, sister with alopecia areata	No	NA	Splenomegaly with lymphadenopathy	Alopecia areata, acneiform rash of face and dorsum of hands	Poly arthropathy	Severe anemia Neutropenia Thrombocytopenia	Recurrent respiratory tract infection, diarrhea
6	Zhang et al., 2021 [17]	6 years	8	F	NA	Yes	No	Splenomegaly with lymphadenopathy	Systemic rash, pyoderma gangrenosum	No	Pancytopenia	NO HTAP, angina with swollen and painful jaw with dysphagia (Ludwig’s angina)
7	Zhang et al., 2021 [17]	10 years	13	F	NA	No	Yes	Splenomegaly	Pyoderma gangrenosum	Pyogenic arthritis	Anemia Neutropenia	No
8	Zhang et al., 2021 [17]	13 years	19	M	NA	Yes	No	Hepatosplenomegaly with lymphadenopathy	No	Pyogenic arthritis	Pancytopenia	Severe HTAP with right heart failure
9	Xu et al., 2021 [18]	3 years	8	F	Healthy parents, non-consanguineous	No	No	Hepatosplenomegaly	No	Unsymmetrical arthritis, aseptic necrosis of femoral head	Anemia Neutropenia	Recurrent infection
10	Xu et al., 2021 [18]	6 years	2.5	F	Healthy parents, non-consanguineous	Yes	No	Hepatosplenomegaly	Yes, eyelid lesions	No	Anemia Neutropenia	Severe pulmonary infection with features of MAS
11	Huang et al., 2021 [21]	13 years	44	M	Healthy non-consanguineous parents	No	NA	NA	Acne, cystic acne, facial papulonodular and hand ulcers	Intermittent elbow pain	Leucopenia	No
12	Huang et al., 2021 [21]	17 years	21	M	Father with PAMI syndrome	No	NA	NA	Cystic acne	Elbow pain	Leucopenia	No
13	Mendonca et al., 2021 [19]	First months	4	F	Healthy parents, non-consanguineous, German–Italian descent	Yes, recurrent	Yes	Splenomegaly	Macular skin rash	Arthritis	Anemia Neutropenia Thrombocytopenia	Diarrhea, severe CMV infection
14	Mendonca et al., 2021 [19]	NA	adulthood	F	NA	No	No	No	No	No	Anemia Neutropenia	During childhood recurrent fever and thrombocytopenia
14	Laberko et al., 2021 [20]	At birth	11.9	F	Russia	Yes	Yes	Hepatosplenomegaly with lymphadenopathy	Soft tissue abscesses, pyoderma, chronic gingivitis, aphtosus, stomatitis	No	Anemia Neutropenia Thrombocytopenia	Chronic gingivitis, stomatitis, X-chromosome derivate (46, X, i(Xp))
16	Laberko et al., 2021 [20]	At birth	1	M	Russia	Yes	Yes	Hepatosplenomegaly with lymphadenopathy	Vasculitis, panniculitis	Arthritis	Anemia Neutropenia Thrombocytopenia	Non-active colitis, ulcerative gastritis, MAS, myocarditis
17	Laberko et al., 2021 [20]	2.9 years	7	F	Russia	Yes	No	Hepatosplenomegaly with lymphadenopathy	Polymorphic rash, vasculitis	Arthralgia, aseptic osteomyelitis	Anemia Neutropenia Thrombocytopenia	MAS
18	Laberko et al., 2021 [20]	2.5 months	0.5	F	Germany	Yes	Yes	Lymphadenopathy	No	No	Anemia Neutropenia Thrombocytopenia	No
19	Del Borrello et al., 2021 [22]	3 months	1.5	M	Negative family history	Yes, recurrent	Yes	Hepatosplenomegaly with lymphadenopathy	Urticarial rash	No	Anemia	Hypotonia, development delay, dystrophic and dysmorphic features, inguinal hernia, axial hypotonia
20	Borgia et al., 2021 [23]	4 years	8	M	Italian, negative family history	NA	Yes	Hepatosplenomegaly	Pyoderma gangrenosum, cystic acne, poor wound healing	Asymmetrical polyarthritis	Anemia Leucopenia Neutropenia	Kidney involvement with focal segmental glomerulosclerosis, growth hormone deficiency
21	Resende et al., 2019 [24]	Childhood	20	F	NA	NA	Yes	Hepatosplenomegaly	Pyoderma gangrenosum	Arthritis	Pancytopenia	Osteoporosis, portal hypertension with esophageal varices, abdominal pain with recurrent diarrhea in childhood
22	Hashmi et al., 2019 [25]	2 years	12	F	Grandfather with telomer biology disorder, no PAID in family	Yes, recurrent	NA	Splenomegaly with lymphadenopathy and lymphadenitis	No	Polyarthralgia	Anemia Leucopenia Neutropenia	Chest pain
23	Hashmi et al., 2019 [25]	At birth	7	F	NA	Yes	Yes	Hepatosplenomegaly with lymphadenopathy	No	Arthralgia	Pancytopenia Epistaxis	Born prematurely at 28 W, RSV infection, staph aureus osteomyelitis, periodontal disease, developmental delay
24	Dai et al., 2019 [26]	18 years	56	F	NA	NA	NA	Splenomegaly	Poor wound healing and wound dehiscence	Symmetrical deforming non-erosive polyarthritis since the age of 18	Pancytopenia	Macronodular cirrhosis, mild portal hypertension, recurrent childhood chest infections, podocyte effacement and glomerular calprotectin dense deposits
25	Mejbri et al., 2019 [13]*	5 months	2	F	Caucasian healthy parents, non-consanguineous, same mutation for the father but no symptoms	Yes	No	Hepatosplenomegaly	No	Multifocal osteomyelitis	Anemia Neutropenia Epistaxis	No
26	Klotgen et al., 2018 [27]	Childhood	23	M	NA	Yes	NA	Hepatosplenomegaly	Pyoderma gangrenosum, skin ulceration, nodulocystic acne	Relapsing arthritis and osteomyelitis	Pancytopenia	Recurrent infection
27	Belelli et al., 2017 [2]	7 years	8	M	mother with PAMI sd	No	No	Hepatosplenomegaly	No	Recurrent right knee swelling	Pancytopenia	No
28	Belelli et al., 2017 [2]	Childhood	Adulthood	F	Son with PAMI sd	No	No	NA	Psoriasis, acne	Arthralgia	Anemia Leucopenia	No history of infections
29	Lindwall et al., 2015 [28]	4 years	25	M	Mother with psoriasic arthritis, no other illness	No	No	Hepatosplenomegaly	Cystic acne, scalp pyoderma, recurrent and multiple pyoderma gangrenosum	Recurrent symmetrical arthritis and hyper-mobile joint, osteomyelitis	Anemia Neutropenia Epistaxis	Septic shock with acute renal and respiratory failure, colitis, cellulitis, acute cholecystitis. Allergies to sulfonamides, ciprofloxacin, vancomycin, and Rocephin
30	Sologuren et al., 2014 [30]	First months	NA	M	NA	Yes	Yes	Hepatosplenomegaly with lymphadenopathy	Abscess	No	Anemia Neutropenia Thrombocytopenia	Hepatic abscess by E. coli
31	Sampson et al., 2002 [8]	6 years	Adulthood	M	NA	No	No	Hepatosplenomegaly	PG, ulcerative dermatitis, furunculosis and pustulosis	Arthritis	Pancytopenia	Liver failure with cirrhosis minimal change Glomerulonephritis, deceased post-op complication at 35 y (liver transplantation)
32	Fessatou et al., 2005 [7]	14 months	11	F	NA	No	Yes	Hepatosplenomegaly with lymphadenopathy	No	No	Anemia Neutropenia Leucopenia	NA
33	Isidor et al., 2009 [5]	11 months	8	M	Non-consanguineous healthy parents	Yes	Yes	Hepatosplenomegaly	Macular erythematous rash, necrotic palpebral lesions	Arthritis, chronic polyarthralgia	Anemia Neutropenia Leucopenia Epistaxis and hematomas	Generalized muscular atrophy and delayed motor development
34	Sugiura et al., 2006 [6]	4 years	20	F	NA	NA	Yes	Hepatosplenomegaly	Pyoderma gangrenosum, ulcerative dermatitis	Arthritis	Pancytopenia	Portal hypertension with ascites and esophageal varix, moderately impaired mental and motor development
35	Holzinger et al., 2015 [3]	First month	5	M	NA	Yes	Yes	Hepatosplenomegaly with lymphadenopathy	Recurrent perianal and gluteal abscesses	No	Anemia Neutropenia Thrombocytopenia	NA
36	Holzinger et al., 2015 [3]	7 months	9	F	Paternal family history of multiple members with early gout, father at 14 y	No	Yes	Hepatosplenomegaly with lymphadenopathy	Pustular lesions	Arthralgia	Pancytopenia	Ig A nephropathy with hematuria and proteinuria (7 y), recurrent infection with cellulitis, conjunctivitis, pneumonia and central line infections, poor weight gain and requirement of gastrotomy tube feeding and growth retardation
37	Holzinger et al., 2015 [3]	4 years	5	M	Father with PAMI sd	No	NA	No	No	No	Anemia Neutropenia	No recurrent infection
38	Holzinger et al., 2015 [3]	18 years	24	M	NA	No	NA	No	Skin abscesses, Acne	Morning stiffness	Anemia Neutropenia/Leucopenia	History of salmonella meningitis at age of 16 m, no recurrent infection
39	Holzinger et al., 2015 [3]	10 weeks	7	F	NA	Yes, recurrent fever	Yes	Splenomegaly	Erythema multiforme rash	Arthritis, transient non infectious osteitis	Pancytopenia	No
40	Demidowich et al., 2012 [9]	1 month	Deceased (36) Sepsis-associated MOF	F	NA	NA	No	Splenomegaly with lymphadenopathy	Pyoderma gangrenosum, bullae, ulcerations, erythematous lesions	Recurrent sterile arthritis	Anemia Neutropeia Leucopenia Bleeding diasthesis	Recurrent upper respiratory infections and pneumonias, episodic lymphangitis and cellulitis, saddle nose deformity following spontaneous septal perforation, pharyngeal papillomatosis, large granular lymphocytosis of the T cells
41	Holzinger et al., 2015 [3]	At birth	5	F	Mother with SLE/RA overlap with Ro/La antibodies	Yes, recurrent	Yes	Hepatosplenomegaly with lymphadenopathy	Eczematous rash, photosensitivity, scalp hair heterochromia	Dactylitis, arthralgia, aseptic osteomyelitis	Pancytopenia	Viral infections, scoliosis, delayed motor and communication development, chronic diarrhea
42	Holzinger et al., 2015 [3]	18 months	16	M	NA	No	Yes	Hepatosplenomegaly with lymphadenopathy	Pyoderma gangrenosum, furunculosis	Symmetric aseptic polyarthritis and aseptic necrosis of the right femoral head	Pancytopenia	Glomerulonephritis (hematuria + Proteinuria)
43	Holzinger et al., 2015 [3]	1 year	17	M	NA	No	No	Hepatosplenomegaly	Acne vulgaris, pyoderma gangrenosum and pustular lesions	Osteomyelitis	Pancytopenia	Bilateral hearing loss after multiple middle ear infections, hepatic noncaseating granulomas, mild steatosis, and mild inflammation

* Complete description of the patient in the Appendix A.

## Data Availability

No new data were created or analyzed in this study. Data sharing is not applicable to this article.

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
