# Peer review of "PSTPIP1-Associated Myeloid-Related Proteinemia Inflammatory (PAMI) Syndrome: A Systematic Review"

_genes, 2023, doi:10.3390/genes14081655_

Round 1

Reviewer 1 Report

In this systematic review the authors provide a comprehensive overview of PAMI syndrome clinical phenotype, disease course, therapy, and outcome. The results are described in a well-structured manner and in adequate detail. Given the rarity of the syndrome, this review contributes to reporting useful conclusions resulting from the analysis of published case reports and series.  There are few minor issues to be addressed:

Comment 1

In the file of supplementary files provided, unfortunately there was no table describing patient characteristics. The content of the archive entitled “PAMI Review patient description” is actually the description of a case. Please provide the relevant table/archive describing patients’ characteristics.

Comment 2

In the manuscript section 3.2 the authors mention: “We found that patients with PAMI syndrome presented with early-onset systemic inflammation along with a variety of auto-inflammatory features ranging from mild autoinflammatory symptoms to severe presentations.” Please consider including in the text a brief description of the main auto-inflammatory features the patients presented.

Comment 3

The authors should consider moving the description of hematological manifestations to the section 3.3 of the manuscript.

Very good quality of English language. Minor editing issues to be addressed.

Author Response

Comment 1

In the file of supplementary files provided, unfortunately there was no table describing patient characteristics. The content of the archive entitled “PAMI Review patient description” is actually the description of a case. Please provide the relevant table/archive describing patients’ characteristics.

A: Table 1 included in the main text with patients’ characteristics.

Comment 2

In the manuscript section 3.2 the authors mention: “We found that patients with PAMI syndrome presented with early-onset systemic inflammation along with a variety of auto-inflammatory features ranging from mild autoinflammatory symptoms to severe presentations.” Please consider including in the text a brief description of the main auto-inflammatory features the patients presented.

A: A brief description has been added (point 3.2).

Comment 3

The authors should consider moving the description of hematological manifestations to the section 3.3 of the manuscript.

A: We adapted the text as suggested.

Reviewer 2 Report

Very nice and easy-to-read review of PAMI patients. 

It would be interesting to discuss also the possibility of discovering new PAMI variants > for example : N236K (see infevers registry)

Why was the case of lupus secondary to PAMI not included? PMID: 34559261 > it would be interesting to suggest including PSTPIP1 in the lupus panels ?

I did not find the Table 2 "The demographics and the clinical presentations of the patients are summarized in Table 2 in the supplementary materials."

I did not understand wether or not the case report named (PAMI Review patient description) in supplemental material was included in the study.

Best regards

Author Response

Comment 1

It would be interesting to discuss also the possibility of discovering new PAMI variants > for example: N236K (see infevers registry)

A: The new variant was not included due to the selection criteria predefined. We added a comment in the discussion as suggested.

Comment 2

Why was the case of lupus secondary to PAMI not included? PMID: 34559261 > it would be interesting to suggest including PSTPIP1 in the lupus panels?

A: This case was included in our review (P?, Ref 17); the case reported by Su et al (PMID 34559261) is the same reported by Zhang et al. We added a comment in the discussion accordingly.

Comment 3

I did not find the Table 2 "The demographics and the clinical presentations of the patients are summarized in Table 2 in the supplementary materials."

A: Table 1 with patients’ characteristics was included in the main text.

Comment 4

I did not understand whether the case report named (PAMI Review patient description) in supplemental material was included in the study.

A: Yes, the patient with the detailed clinical description is included in our review, it is the same reported by Mejbri et al (ref 13).